# Genomic and Proteomic Analysis of Six Vi01-like Phages Reveals Wide Host Range and Multiple Tail Spike Proteins

**DOI:** 10.3390/v16020289

**Published:** 2024-02-13

**Authors:** Evan B. Harris, Kenneth K. K. Ewool, Lucy C. Bowden, Jonatan Fierro, Daniel Johnson, McKay Meinzer, Sadie Tayler, Julianne H. Grose

**Affiliations:** Department of Microbiology and Molecular Biology, Brigham Young University, Provo, UT 84604, USA; harrisevan715@gmail.com (E.B.H.); kewool@student.byu.edu (K.K.K.E.);

**Keywords:** Vi01, bacteriophage, *Enterobacteriaceae*, tail spike protein, cluster, phage therapy

## Abstract

*Enterobacteriaceae* is a large family of Gram-negative bacteria composed of many pathogens, including *Salmonella* and *Shigella*. Here, we characterize six bacteriophages that infect *Enterobacteriaceae,* which were isolated from wastewater plants in the Wasatch front (Utah, United States). These phages are highly similar to the *Kuttervirus* vB_SenM_Vi01 (Vi01), which was isolated using wastewater from Kiel, Germany. The phages vary little in genome size and are between 157 kb and 164 kb, which is consistent with the sizes of other phages in the Vi01-like phage family. These six phages were characterized through genomic and proteomic comparison, mass spectrometry, and both laboratory and clinical host range studies. While their proteomes are largely unstudied, mass spectrometry analysis confirmed the production of five hypothetical proteins, several of which unveiled a potential operon that suggests a ferritin-mediated entry system on the Vi01-like phage family tail. However, no dependence on this pathway was observed for the single host tested herein. While unable to infect every genus of *Enterobacteriaceae* tested, these phages are extraordinarily broad ranged, with several demonstrating the ability to infect *Salmonella enterica* and *Citrobacter freundii* strains with generally high efficiency, as well as several clinical *Salmonella enterica* isolates, most likely due to their multiple tail fibers.

## 1. Introduction

Bacteriophages (phages) are the most common and diverse biological entity in the world, with some estimates bringing the number of virions to 10^31^–10^32^ [1,2,3,4]. Phages have strong antibiotic capabilities, being natural predators of bacteria. During the lytic infection of their host bacteria, phages insert their foreign DNA into the host cell, ending with phage replication and assembly followed by release through cell lysis and death [5,6]. In addition, some phages integrate into the host genome as “temperate” phages and are replicated with the host DNA and thus contribute to host functions and evolution. The abundance of phages, the relative ease of phage discovery, and their clear influence on the evolutionary pathways of bacteria provide great insight into the ecology and evolution of bacteria [7]. This insight is essential to understanding and treating the threat of multidrug resistant bacterial strains [8,9].

*Enterobacteriaceae* is a large family of Gram-negative bacteria, first classified in the 1930s [10]. It is composed of many pathogens, including *Salmonella*, *Enterobacter*, *Citrobacter*, *Shigella*, *Proteus*, *Serratia*, *Klebsiella*, *Escherichia coli*, and others. They are bacilli, typically between 1–5 μm in length, do not form spores, and may be either motile or nonmotile. They are often normal members of the gut microbiome but pose serious risks when present in other areas of the body. They may cause urinary tract, intestinal, and blood infections [11]. *Enterobacteriaceae* have additionally been frighteningly efficient at developing antibiotic resistance via mutation or plasmid-mediated whole gene acquisition [12]. The American Journal of Medicine reports that as of 2006, approximately 20% of *Klebsiella pneumoniae* infections and 31% of *Enterobacter* spp. infections in American ICUs are not susceptible to third generation cephalosporins [13]. These numbers have only been increasing in the subsequent years [14,15]. Clearly a deeper understanding of *Enterobacteriaceae* and the phages that infect them is imperative to human health, both to increase the understanding of host evolution and treatment options.

We have recently discovered a total of six *Enterobacteriaceae* phages, namely *Salmonella* typhimurium phages Guerrero, AR2819, FrontPhageNews, SilasIsHot, and Sajous1, as well as one *Shigella* phage ChubbyThor, each of which logged characteristics of the Vi01-like (AKA Viuna-like) phage family (Table 1). This family was proposed in 2010, when *Salmonella* typhimurium-specific phage Vi01 was proposed as a new lineage of Myoviridae [16] and has since been designated as part of the *Ackermannviridae* family by the International Committee on Taxonomy of Viruses (ICTV) [17]. AR2819, FrontPhageNews, ChubbyThor, Sajous1, and SilasIsHot have been previously published in a genome announcement [18] while the isolation, sequencing and initial characterization of Guerrero is described herein. Basic genomic characteristics of these phages can be found in Table 1.

Vi01, like the second phage discovered in this family SboMAG3, was found to be highly specific to its preferred host, which is theorized to be the case due to a virulence capsule antigen-degrading acetyl esterase domain found to be incorporated into one of the phage’s three tail spikes [16]. Since 2010, many more phages have been classified as Vi01-like and as of October 2023, we were able to identify a total of 150 Vi01-like phages that infect *Enterobacteriaceae* deposited in NCBI GenBank. Seventy of these Vi01-like phages appear in scientific articles in PubMed, primarily in genome announcements [19,20,21,22,23,24,25,26,27,28,29,30,31,32,33,34,35,36,37,38,39,40,41,42,43,44,45,46,47,48,49,50,51,52,53,54,55,56,57,58,59,60,61,62,63,64,65,66,67,68,69,70,71,72,73,74,75,76,77,78,79,80,81,82]. Despite their apparent ease in isolation and widespread nature across several bacterial hosts, few papers have focused on the characterization of these phages (Appendix A). Herein we present a broad genomic comparison of the phages in this family as well as further characterization of a representative phage through mass spectrometry. In addition, we explore their potential use in phage therapy through host range studies of five of these new phage isolates.

## 2. Materials and Methods

### 2.1. Phage Isolation and Host Range

Bacteriophage Guerrero was isolated from wastewater through LB-based enrichment culture grown at 37 °C for 48 h. Bacteria were pelleted by centrifugation and the supernatant was incubated with fresh bacterial overnight for 30 min. and plated for plaques on LB top agar. A single plaque was purified by once again incubating with fresh bacterial overnight cultures and plating in LB top agar. This single plaque isolation and reinfection was repeated three times. A lysate (>10^8^ plaque forming units/mL) was made by incubating a plaque from the final purification plate with diluted bacterial overnight in LB. Host range experiments were conducted with high titer lysates (>10^8^ pfu/mL) and were performed using spot assays to identify positives and negatives, followed by plaque assays to determine efficiency. Briefly, 5 uL of phage lysate was spotted onto 0.5 mL bacterial overnight that was plated in LB top agar. All negatives were confirmed through three independent high titer spot assays, while any positives were confirmed through efficiency plaque assay in a minimum of two independent assays. Plaque assay consisted of incubating 10-fold dilutions with 0.5 mL of bacteria for 30–45 min, followed by plating in LB top agar.

### 2.2. Genome Analysis 

The Genomic DNA of phage Guerrero was isolated with the Norgen Biotek Phage DNA Isolation Kit (Thorold, ON, Canada) and was prepared for paired-end Illumina HiSeq 2500 sequencing with the New England Biolabs (New England Biolabs, Ipswich, MA, USA) Ultra II DNA kit. Geneious version R.11 [83] was used to assemble the genome, which circularized upon assembly and was subsequently annotated using DNA Master version 5.23.6 [84] and GeneMarkS [85]. All software was used at default settings. The additional genomes included in this study were obtained through NCBI GenBank (see Appendix A). Whole genome nucleotide dot plots were constructed using Genome Pair Rapid Dotter (Gepard) [86]. A phylogenetic tree was produced using the Mega11 software using the Neighbor-Joining Method [87,88]. The core genome was identified using CoreGenes 3.5 at default settings [89]. Genome homology was visualized using Clinker [90] and Roary [91]. Average Nucleotide Identity (ANI) analysis was performed using FastANI at default settings [92].

### 2.3. Electron Microscopy 

Samples for SEM analysis were prepared by placing 15 μL of high-titer bacteriophage lysate on a 200-mesh copper carbon type-B electron microscope grid for one–two minutes. The lysate was wicked away and the grids were stained for 2 min using 15 μL of 2% phosphotungstic acid (pH = 7) or uranyl acetate. Residual liquid was wicked away using Kimtech wipes and the grid was allowed to dry before being imaged. Electron microscopy was performed at Brigham Young University in the Life Sciences Microscopy Lab using ana FEI Helios NATOCAB 600i DualBeam FIB/SEM microscope with STEM detector.

### 2.4. Mass Spectrometry 

Two liters of Sajous1 phage lysate were centrifuged at 12,000× *g* for 15 min at 4 °C. The supernatant was discarded and DNase I and Rnase A were added to the supernatant to a final concentration of 1 µg/mL each. The phage solution was concentrated by pelleting the phages using a Sorvall GSA centrifuge at 7000× *g* for 18 h at 4 °C. The phage pellets were resuspended in SM buffer and centrifuged again at 12,000× *g* for 10 min at 4 °C to remove any remaining cell debris. A CsCl gradient was created using 0.75 g CsCl per ml of phage suspension. Using a Sorvall GSA rotor, the mixture was centrifuged at 25,000 RPM for 24 h at 5 °C. The phage band was pulled and transferred to a dialysis cassette, which was placed in 1 L of gelatin-free SM buffer (50 mM Tris-HCl, 8 mM magnesium sulfate, pH 7.5) containing 1 M NaCl at 4 °C overnight. The cassette was then transferred to 1 L of standard gelatin-free SM buffer (containing 0.1 M NaCl) for 2–3 h at room temperature, which was immediately repeated. The phage was then trypsin digested and prepared for liquid chromatography with tandem mass spectrometry (LC-MS-MS) using Fastprep (MP BioMedicals, Irvine, CA, USA). The spectra identified were mapped back to the genome by BLASTP.

## 3. Results

### 3.1. Analysis of Six Vi01-like Phages and Their Replationship to 144 Vi01-like Enterobacteriacae of the Ackermannviridae

#### 3.1.1. FrontPhageNews, Guerrero, Sajous1, SilasIsHot, AR2819, and ChubbyThor Lie in Two of Five *Enterobacteriaceae Ackermannviridae* Subclusters

Phages are incredibly diverse and lack a common homologous gene, making a single phylogenetic tree impossible [93]. Because of this, one descriptive way that phages are grouped is phage families called “clusters”. Phages in these families are typically defined as sharing greater than 50% of the homology of their genome, allowing newly discovered phages to be easily classified [94,95]. In order to obtain a full picture of the breadth of the Vi01-like phage cluster first described by Casjens and Grose [4] which are members of the *Ackermannviridae* phage family designated by the ICTV [17], NCBI GenBank was searched for phages with major capsid protein similarity of greater than 80% and as of October 2023, 150 fully sequenced *Enterobacteriaceae* phages were identified as possible cluster members and subsequently confirmed by Gepard dot plot analysis (Appendix A). Figure 1 contains a Gepard dot plot analysis [86] of 61 representative phages from the 150 identified (the additional phages are highly similar to phages appearing in this dot plot; however, only 61 could be graphed at once). Dot plot comparison reveals five *Enterobacteriaceae Ackermannviridae* subclusters (A, B, C, D, and E) that have similarity over 50% of the genome, with *Salmonella* phages FrontPhageNews, Guerrero, Sajous1, SilasIsHot and AR2819 in cluster A and *Shigella* phage ChubbyThor in cluster B (Figure 1a). This is one additional subcluster from the four reported by Grose and Casjens in 2014 from analysis of only 16 reported Vi01-like *Enterobacteriaceae* phages [4]. Electron microscope analysis of phages Guerrero and AR2819 verified the morphologic resemblance to Vi01-like Myoviridae (Figure 1b,c).

A survey of the relative hosts associated with each subcluster reveals a clear relationship of host with subcluster (Figure 1a) and all six of our phages follow this rule. There are only two exceptions among the 61 analyzed by dot plot where host is not directly correlated with subcluster designation (a single phage exception in subcluster B as well as in subcluster C); however, analysis of all 150 *Enterobacteriaceae Ackermannviridae* gives a clearer picture. Of the 150 phages, subcluster A comprises phages of only *Salmonella* or *E. coli Enterobacteriaceae* hosts (96 phages) and has been designated the *Kuttervirus* genus by the ICTV [17]. In contrast subcluster B contains 16 phages of diverse hosts namely *Salmonella*, *Shigella*, *E. coli* or *Enterobacter* corresponding to the *Agtrevirus* ICTV genus, as well as 16 phages with *Dickeya* hosts that are known as the *Limestonevirus* ICTV genus. This is the one case where our dot plot subcluster analysis differs substantially from the ICTV genus assignments where two ICTV genera exist in one subcluster. Subcluster C (17 total phages corresponding to the ICTV *Taipeivirus* genus) consists of predominantly *Klebsiella* phages and a single *Serratia* phage as well as a single *E. coli* phage, while two *Serratia* phages are the only members of cluster D (known as the *Miltonvirus* ICTV genus). Cluster E contains two *Erwinia* phages and is known as the *Nezavisimistyvirus* ICTV genus. Of note, there are other *Ackermannviridae* genera proposed with only members that infect non-*Enterobacteriaceae* hosts that are not analyzed in this manuscript, namely *Vibrio* phages (the *Vapseptimavirus* and *Kujavirus* ICTV genera) and an *Aeromonas* phage (the *Tedavirus* genus), displaying the wide-spread success of the *Ackermannviridae* phage family. Our subcluster designations are used throughout the remaining manuscript (subclusters A–E).

Measurements were taken of the phages using existing electron microscope images and compared with measurements found in previous studies, where present [20,35,69,73,93,94]. The phages appear to be of a similar size, with reported averages of capsid (85 ± 15 nm) and tail (125 ± 20 nm) within 20% as expected for similar genome size, but tail width (19 ± 5.6 nm) and neck (14 ± 15.5 nm) being more variable as seen in Table 2.

#### 3.1.2. Characteristics of Representative Phages of Vi01-like Subclusters

Due to the high genomic similarity within each subcluster, one phage was selected from each subcluster to be the basis of further genomic comparison: namely Vi01 (A), ChubbyThor (B), Magnus (C), 3M (D), and Bue1 (E) [19,73,94]. A summary of the characteristics of these phages is provided in Table 3.

The average nucleotide identity within subclusters is high (>90%) while the ANI between the five subclusters is 61% and 73% with these representative phages, as seen in Table 4. Subclusters A, B, and C are most similar to one another, with an ANI of ~72%. Subclusters D and E are the most dissimilar, sitting between roughly 61% and 64% similarity to every other subcluster including one another. They stand apart from the high similarity group described above, but also are equally different from one another. This is likely due to their hosts which are *Serratia* and *Erwinia*, which are more dissimilar to all other hosts than are *E. coli*, *Salmonella*, *Shigella* and *Klebsiella*.

A phylogenetic tree was produced using the major capsid protein of 150 of the Vi01-like *Enterobacteriaceae* phages on NCBI [87,96,97,98,99]. This is represented in Figure 2, which has been labeled using the subclusters (A–E) identified by the dot plot and ANI analysis above. Phage MCPs appeared to be most similar to orthologous genes in their same subclusters, verifying the subcluster designations made by dot plot and ANI and suggesting little homologous recombination of MCP’s between subcluster, perhaps due to the differences between hosts associated with each subcluster [16]. The MCP phylogenetic tree also suggests that the A,B,C evolved from a common progenitor and diverged from D,E, which has a more shallow branch point.

#### 3.1.3. Analysis of the Conserved Proteins among the Subclusters of the Vi01-like *Enterobacteriaceae*

Using the CoreGenes program at default settings, we were able to identify the core genes present across each of the five subclusters identified above using the one representative from each subcluster along with FrontPhageNews to investigate intra-subcluster relatedness to Vi01 (Table 5) [89]. Phage Vi01 was used as the reference genome in this analysis of six phages in total and 114 core genes were identified. The core genome of the Vi01-like family contains 21 structural protein encoding genes, 41 DNA/RNA-associated protein encoding genes, 3 lysis protein encoding genes, 7 phage assembly protein encoding genes, 3 virulence genes, and 29 hypothetical protein coding genes. In total, the core 114 gene Vi01-like phage genome ranges between 55–67% of the size of the genomes of each phage in the cluster. That is, over half of the genome by gene count is conserved between each representative of each subcluster.

These core genes appear to be fairly interspersed throughout the genomes, with structural and assembly genes as well as those involved in DNA/RNA pathways being dispersed throughout the genome. The order of the genes, however, is mostly conserved between the representative phages. There are occasions where gene synteny is broken and genes appear to be randomly distributed throughout the genome as seen in Figure 3. Often, this manifests in small, ~500 bp proteins that, as of yet, have not been defined in any species. Vi01 gp187, ChubbyThor gp18, Magnus gp20, 3M gp67, and FrontPhageNews gp21 are all highly conserved, though uncharacterized. These gene products are situated roughly 1 kb upstream of a DNA polymerase with the exception of 3M gp67. Rather than sitting just upstream of the DNA polymerase, it can be found 68 kb upstream. This seems to be a result of some type of recombination. Besides a handful of outliers, however, the genomes share a remarkable similarity of genome composition; the structure and placement of genes within the genome is highly conserved between these inter-cluster groups. Among these conserved similarities are important structural proteins like the baseplate hub subunit and portal protein, vital enzymes like endonuclease and primase, RIIA and RIIB lysis inhibitors, and DNA binding proteins like DNA ligase and DNA polymerase clamp loaders.

One notable area of difference between the phages is the area beginning between approximately 145 kb and 157 kb which is a segment of the genome that contains structural proteins. Specifically, they are tail proteins. This area contains the least amount of synteny between the phages. Phage tail proteins are one of the main determinants of host specificity, so although the phages are highly conserved this is the area in which we would expect to see the most divergence [100]. This effect is also seen at the structural areas noted at 60 kb and 90 kb, which both code for tail proteins and are also areas of low conservation.

Another notable difference is the presence of nicotinate phosphoribosyl transferase and ribose phosphate pyrophosphokinase, which are important enzymes for the biosynthesis of NAD(+) and phosphoribosyl pyrophosphate, respectively. In the examined phages, these protein-coding sequences were only found in phages ChubbyThor (subcluster B) and Magnus (subcluster C), in which the sequences were highly conserved (97.18% identity) and in the same location (~117.5 kb). These proteins were not found in the other subcluster representatives (even by a tBLASTN search to detect mis-annotation), but an NCBI protein sequence blast revealed these proteins in other cluster B and C phages and also in some phages from cluster A (such as *Salmonella* phage Allotria), whereas Vi01 itself does not encode it, suggesting they were either lost in some members of the subclusters or acquired by others.

### 3.2. Proteome Characterization of Vi01-like Phages through Structural and Operon Analysis, as Well as Mass Spectrometry

#### 3.2.1. Structural and Operon Analysis of a *Salmonella* phage (FrontPhageNews) of Cluster A and a *Shigella* Phage (ChubbyThor) of Cluster B Provides Putative Functions for 45 Hypothetical Proteins

As it stands, proteomic analysis of most phage contains what has been classified as a large amount of proteomic ‘dark matter’, or proteins with unknown function [101], and the 150 Vi01-like phages analyzed herein are no exception, with most fully annotated phages harboring 50–60% hypothetical or uncharacterized proteins, making further protein analysis imperative in understanding them. Herein the hypothetical and uncharacterized proteins of two phages, a *Shigella* phage (ChubbyThor) of cluster B and a *Salmonella* phage (FrontPhageNews) of cluster A, were analyzed looking for structural homology and putative functions based on protein folding trends, included in Appendix A. In total, high (>70%)-confidence structural homologs for 37 hypothetical proteins encoded in the genome of FrontPhageNews were found using Phyre2 [102], positing putative functions for these unknown proteins while a total of 26 were found for ChubbyThor. These proteins were then analyzed by HHPRED to check the validity of the results and a majority were supported (Appendix A). Several of these proteins suggest novel pathways for the Vi01-like phages. Of particular interest, gp98 of FrontPhageNews was found to share 39% structural alignment with the Human C complex spliceosome with a 72.2% confidence, suggesting that RNA splicing could be occurring in this phage and others in the Vi01 family. ChubbyThor’s gp160 shared 26% structural alignment with FtsX with 77.1% confidence. FtsX is a part of the FtsEX complex in *Streptococcus pneumoniae*, a membrane bound complex that transports proteins utilized in cell division [103]. Disruption of the cell division pathways has been previously reported to facilitate phage replication by allowing for cell elongation.

Operon analysis is an additional tool for protein function prediction, in that proteins within an operon usually have related function [104]. Visualized in Figure 4, we identified several likely operons containing proteins of unknown function [105]. Operons were predicted using the Operon-mapper software at default settings [106]. Hypothetical proteins gp173 and 174 from FrontPhageNews are within an operon that includes genes for a translational repressor protein, a DNA polymerase clamp holder, and a clamp loader subunit. These genes could therefore be associated with a DNA polymerase. Given the presence of the repressor gene, they could play a role in inhibiting the synthesis of this polymerase. Gp199 is contained in an operon consisting of genes for head structural proteins and may, therefore, be an additional protein contributing to the structure of the phage capsid. In ChubbyThor, gp43, 44, and 45 are found within an operon containing a transposase, a ribonuclease, and a DNA binding protein, so these may also be involved in transposon activity. Gp67 and 69 are within an operon containing a DNA repair purposed ATPase and a deaminase, so they may also encode DNA repair proteins. Together, structural and operon analysis provide putative functions (or at least pathways) for 45 proteins.

#### 3.2.2. Mass Spectrometry Analysis of Sajous1 Identifies Putative Virion Proteins

Mass spectrometry analysis of cesium chloride (CsCl) purified phage Sajous1 was able to identify spectra corresponding to 31 previously predicted structural proteins, 9 DNA-associated proteins, 2 cell lysis proteins, 5 phage assembly proteins, 6 proteins of miscellaneous function, and 8 hypothetical proteins that are now known to be expressed and are likely part of the virion (Table 6). Gene products 157, 136, 139, 140, and 214 were the top five most highly represented. They were determined to be the major capsid protein, a tail fiber protein, an exo-alpha-sialidase (likely involved in cell wall degradation for phage entry) a putative virulence-associated VriC protein previously associated with host range, and a tail completion protein respectively [47]. Proteins with high counts of retrieved spectra are likely part of the virion, but some with few peptide counts may be contaminants from cellular debris despite purification. However, known virion-associated proteins such as gp142 (a previously reported structural protein) and gp154 (a prohead core protein) also have low peptide counts, making a strict low peptide count unable to distinguish between virion proteins and cell debris contaminants.

Using Phyre2 analysis, we were able to identify putative structures for the noted hypothetical proteins, shown in Table 7. Gene products 215 and 217 have high confidence values (>60% confidence) for their putative functions, a molybdate binding domain protein and Ferritin, respectively. Gene products 29, 153, and 216 have lower confidence values, but appear to be related to methionine synthase, ATP dependent helicase, and the zeta-subunit of DNA polymerase.

Combining information gathered from mass spectrometry and comparing it to the putative gene functions identified during annotation, we hypothesize that there may be a polycistronic operon encompassing gene products 214–217. These genes are, respectively, a tail completion protein, a molybdate-binding domain protein, an ATP dependent DNA helicase, and ferritin. It has been proposed that phages may use a strategy called the ‘ferrojan horse’ method of bacterial cell wall attachment for entry [107]. According to this hypothesis, phages hide iron and molybdate ions within tail fibers to try to utilize the bacterial cell’s ion uptake pathways. As these gene products are found in the phage proteome, it is likely that this is a strategy that is in use by the phage.

### 3.3. Host Range Analysis of Five Enterobacteriaceae Vi01-like Phages

#### 3.3.1. Host Range of Iron-Uptake Mutant Strains and Common Laboratory *Enterobacteriaceae*

In light of the predicted “ferrojan horse” mode of entry for phage Sajous1, *Salmonella enterica* serovar Typhimurium TonB- and FeoB-deficient strains developed by Tsolis et al. were assessed for phage susceptibility [108]. There are two proteins vital to iron mediated phagocytic infection, TonB and FeoB. FeoB encodes for a homolog of an *E. coli* cytoplasmic membrane iron permease and TonB is an essential element in TonB-dependent siderophore transport. The strains are mutants of *S. enterica* serotype Typhimurium strain IR715 [108,109]. It was found that the five Vi01-like phages had similar infection efficiencies in these mutant strains as compared to the wild-type strains. The data, shown in Table 8, does not appear to suggest the ferrojan horse method to be the only method the phage has for viral infection, at least in the *S. enterica* serovar Typhimurium IR715 host. This suggests that these phages have entry mechanisms that make the proposed ferrojan horse method redundant under laboratory conditions or that they use these anions for some other purpose. This warrants further study of other hosts beyond the scope of this project. The host range analysis of five of the novel Vi01-like phages was also performed on several other common laboratory strains including *S. enterica* serovar Typhimurium, *Citrobacter freundii*, *Cronobacter sakazakii*, *Enterobacter cloacae*, and *Escherichia coli*, *Erwinia amylovora*, *Klebsiella pneumoniae*, *Serratia marcescens,* and *Shigella boydii* (Table 8). In general, the phages were found to have a wide host range capable of infecting multiple genera, with all five tested phages able to infect both *Salmonella typhimurium* and *Citrobacter freundii* lab strains. *Shigella* phage ChubbyThor has the broadest host range in this data, *Shigella boydii* as well as *Salmonella typhimurium* and *Citrobacter freundii*.

#### 3.3.2. Host Range of Clinical *Enterobacteriaceae* Isolates

The five phages were also tested for their ability to infect clinically relevant *Enterobacteriaceae* strains (Table 9). Isolates 0031 and 0409, both *S. enterica* serovar Typhimurium strains, showed plaques from all five phages with *Shigella* phage ChubbyThor displaying reduced efficiency on 0031. Isolate 0404, a *Salmonella heidelberg* strain, showed plaques when combined with every phage excluding FrontPhageNews, while ChubbyThor showed a 10,000-fold reduction on this *Salmonella* strain compared to 0409. *Shigella sonnei* isolates 0422 and 0426 each showed plaques with one phage, Sajous1. In both instances, infection occurred at a reduced efficiency (an approximately 1-million-fold reduction), suggesting these may be mutant phages of some sort. Although Sajous1 was the only phage capable of infecting *E. coli* in laboratory strain culture, our clinical O157:H7 *E. coli* isolates showed plaques when introduced to Sajous1, AR2819, and FrontPhageNews. These results are generally consistent with the host infection efficacies in the nonclinical strains (Table 8) and suggest these phages may be useful in a clinical setting. We see high counts of plaque forming units across the board in *S. enterica* serovar Typhimurium strains. This seems to indicate that amongst the strains of *S. enterica* tested, phage titer concentrations were not affected by the antibacterial resistance mechanisms present in the clinical strains.

#### 3.3.3. Tail Spike Protein Analysis Reveals Four Tail Spike Proteins in Five *Enterobacteriaceae* Vi01-like Phages That May Explain Their Broad Host Range

Recently Sorensen et al. performed an in-silico analysis of 99 *Ackermannviridae* family phages tail spike proteins that suggested a direct correlation with genera (much like the MCP analysis presented in Figure 2) [47]. They found that most *Ackermannviridae* encoded up to four tail spike proteins that fall into four distinct types, which they termed TSP1-TSP4 with several subtypes. The tail spike genes are generally flanked by the conserved virulence associated gene (*vriC*) and a baseplate wedge gene. Sorenson et al. also identified a conserved motif (GTTAVSL) in TSP1, TSP3 and TSP4 of *Kuttervirus* phages that may allow recombination and hence host specificity alterations. A similar comparison of the tail spike proteins of the six phages reported herein is provided in Table 10.

*Salmonella* phages AR2819, FrontPhageNews, Guerrero, SilasIsHot, and Sajous1 each encoded a recognizable TSP1, TSP2, TSP3 and TSP4 that was most closely related to TSPs in the genera to which they belong (the *Kutterviridae*), consistent with the findings of Sorensen et al. (Table 10). However, the only two phages to display the same TSP1 through TSP4 subtypes were Guerrero and Sajous1. Analysis of TSP1 revealed that all four *Salmonella* phages AR2819, FrontPhageNews, Guerrero, Sajous1 and *Shigella* phage ChubbyThor share a TSP1-2 tail fiber, while *Salmonella* phage SilisIsHot has a unique TSP1-18. Analysis of TSP2 revealed that SilasIsHot also contains a unique TSP2. Little is known about the TSP2-5 subtype of SilasIsHot, while the TSP2-1 subtype, which the rest of the phages harbor, has recently been shown to contribute to off-target *Citrobacter* recognition by Gil et al., explaining our *Citrobacter* host range results in Table 9 [110]. In addition, TSP2-1 is known to bind and degrade the O:157O-antigen on Shiga toxin (Stx) producing *E. coli*, which hinted at their ability to infect the O157:H7 clinical strains [111]. Clinical O157:H7 strains were therefore acquired and tested (Table 9). As predicted, only SalisIsHot was unable to infect the O157:H7 isolates. Sajous1 displayed a reproducible reduced efficiency on O157:H7 for unknown reasons (perhaps due to variation in other tail spike proteins or variation in TSP2-1). Analysis of TSP3 revealed all five *Salmonella* phages contain TSP3-1 which is predicted to recognize *S. enterica* serovars including *S*. Typhimurium, *S.* Derby, *S*. 4.12:i:-, *S.* 4.5.12:1:, *S.* enteritidis, and *S.* Goettingen O:4 and O9, explaining why *Salmonella* is a common host [48,110]. Analysis of the final tail fiber, TSP4, revealed FrontPhageNews contains a unique TSP4 among these six phages, which could explain its inability to infect *S.* Heidelberg; however, TSP4-2 has been suggested to recognize *E. coli* O:78 [49].

Conserved *Shigella* phage ChubbyThor encodes three (not four) recognizable tail fibers that were most closely related to those of other *Agtreviridae* (primarily *Shigella* phage AG3 and *Salmonella* phage P46FS4), including two TSP1 proteins and no recognizable TSP3, even when the AG3 TSP3 was used to search the ChubbyThor genome by NCBI’s TBLASTN.

## 4. Discussion

In this study, we report the further characterization of six phages isolated from local wastewater, and their relationship to a large section of the phages identified in the Vi01-like family (part of the ICTV *Ackermannviridae*). The other 144 fully sequenced and annotated *Enterobacteriaceae Ackermannviridae* available on NCBI have been discovered and isolated all over the world. Whole genome analysis of these 150 phages suggests five subclusters of *Enterobacteriaceae Ackermannviridae*, an increase of only one subcluster from a 2014 analysis of only 16 Vi01-like *Enterobacteriaceae Ackermannviridae*, suggesting high relatedness of these phages isolated from around the world. Herein one representative of each subcluster was studied to compare their genomic traits, similarities, and differences. The subclusters, simply titled A–E, were unsurprisingly found to be very similar superficially containing high protein conservation despite the weaker nucleotide conservation (Figure 1, Figure 2 and Figure 3). The representatives from each subcluster had genomes of very similar length, between 157 and 164 kb in length, and each made up of approximately 200 genes. Their GC content was also within a very similar range, the largest difference being about 6%. On average the subclusters rested at about 60–70% average nucleotide similarity to one another.

Despite these differences the *Enterobacteriaceae Ackermannviridae* contain a core genome that was found to be 114 genes, mostly made up of DNA/RNA and structural proteins. In total, the core 114 gene Vi01-like phage genome ranges between 55–67% of the size of the genomes of each phage in the cluster. That is, over half of the genome by gene count is conserved between each representative of each cluster. Clinker analysis also revealed a high amount of similarity in gene distribution (synteny) throughout the genome, with one notable area roughly 12.5 kb in length which clinker identified as dissimilar. This area was populated with genes that encode tail spike and tail fiber proteins, which has been previously reported to be highly variable in *Ackermannviridae*, facilitating broad host ranges among this phage family [49].

Approximately 50–60% of these genomes contained hypothetical proteins of unknown function. Using mass spectrometry, seven hypothetical proteins were validated through expressed spectra. Using Phyre analysis, putative functions were discovered that point to utilization of the hypothesized “ferrojan horse” method of cell entry, where a phage hides iron ions in its tail fibers to attach to the bacterial cell using already existing siderophore iron uptake pathways. However, preliminary investigation revealed that while Vi01-like phages may contain the ability to use this method of entry, it does not appear to be the phage’s main method of infection for the *Salmonella enterica* Typhimurium host investigated herein.

In host range analysis, the phages appeared to hew closely to the hosts in which they were discovered. All phages were able to infect *Salmonella* Typhimurium strains, including lab strains, iron uptake pathway mutant strains, and clinical CRE resistant strains. Of the other clinical strains, *Salmonella* Heidelberg showed phage infection in all tested phages other than FrontPhageNews, which may be due to its unique tail fiber TSP4-2 (see Table 9). These host range findings suggest these phages may be good candidates for *Salmonella* phage therapy and may aid in chimeric design for precise applications [110].

## Figures and Tables

**Figure 1 viruses-16-00289-f001:**
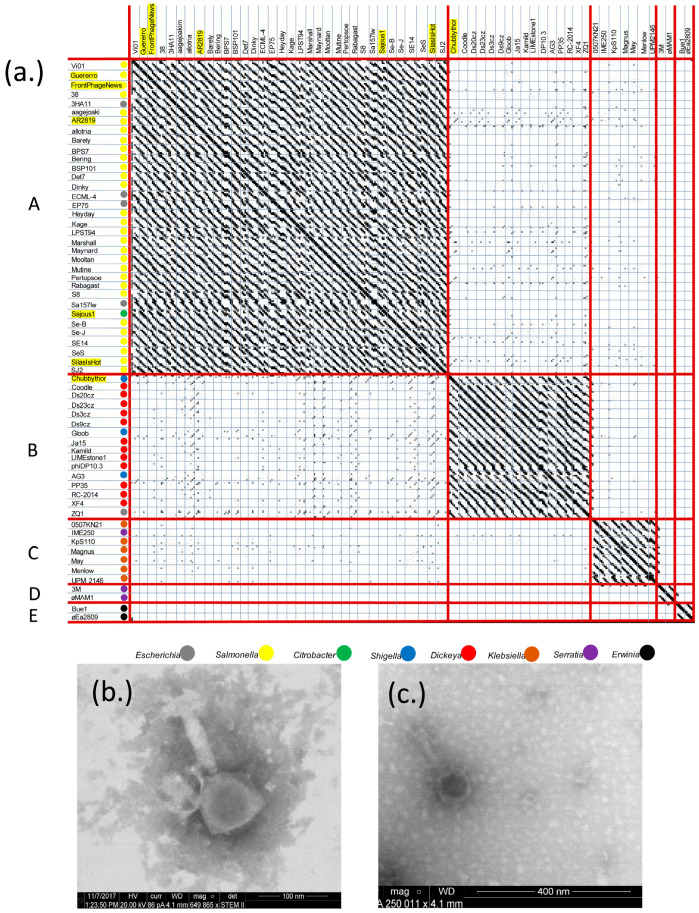
(**a**) Whole genome dot plot analysis of 61 Vi01-like phages reveals five subclusters of related phages within the *Enterobacteriaceae Ackermannviridae* family, with host (indicated in the colored circles) associating with subcluster designation. Yellow highlighted phages were discovered in our lab. The dot plot was constructed using a locally installed version of Gepard at default settings with whole genome nucleotide sequences as the input files. Default settings are defined in Krumsiek et al., 2007 [86]. (**b**) Guerrero as imaged by electron microscopy. (**c**) AR2819 as imaged by electron microscopy.

**Figure 2 viruses-16-00289-f002:**
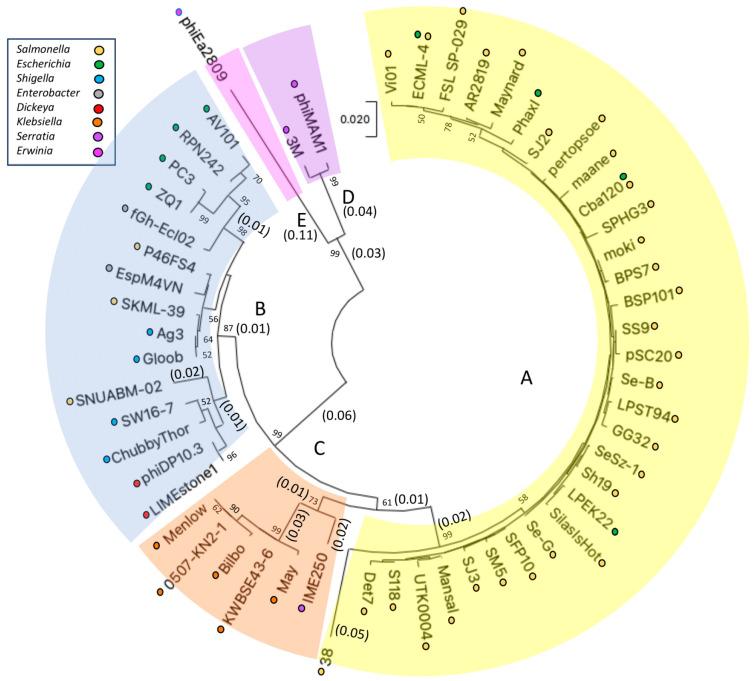
The evolutionary history of 56 unique major capsid proteins (MCPs) identified from 150 *Enterobacteriaceae Ackermannviridae* phages was inferred using the neighbor-joining method [96] and their amino acid sequence. The optimal tree is shown. Phage subcluster designations associated with this manuscript are indicated by branch color and are marked A–E, hosts are indicated by the key, and phages containing identical MCP’s are not shown with the complete phage list is available as Appendix A. The percentage of replicate trees in which the associated taxa clustered together in the bootstrap test (1000 replicates) are shown next to the branches that are >50% (bootstrap tree is not shown) [99]. The tree is drawn to scale, with branch lengths in the same units as those of the evolutionary distances used to infer the phylogenetic tree, with key distances shown in parenthesis as reference. The evolutionary distances were computed using the Poisson correction method [97] and are in the units of the number of amino acid substitutions per site. All ambiguous positions were removed for each sequence pair (pairwise deletion option). There was a total of 447 positions in the final dataset. Original Muscle alignment and subsequent evolutionary analyses were conducted in MEGA11 [87,98].

**Figure 3 viruses-16-00289-f003:**
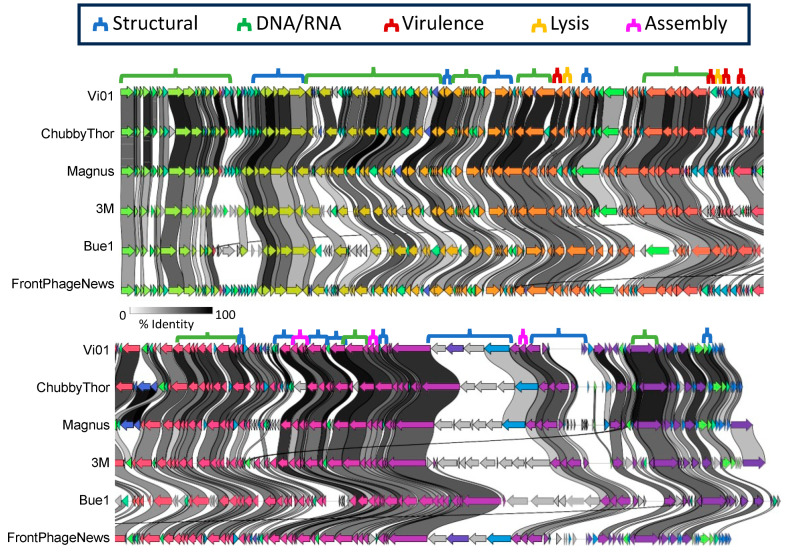
Analysis of the representatives of each Vi01-like *Enterobacteriaceae* subcluster reveals the general homogeneity of their genomes. This figure was made using Clinker [90] at default settings, cut at midpoint, and labeled by hand with identified protein categories. Gene products are indicated by arrows, with direction indicating the encoding strand.

**Figure 4 viruses-16-00289-f004:**
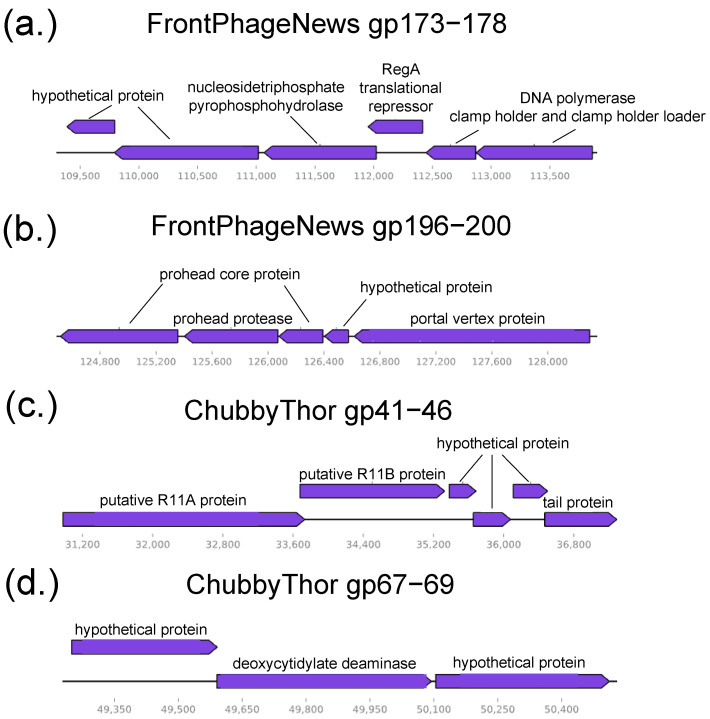
Four operons identified in Vi01-like *Ackermannviridae* phages FrontPhageNews and ChubbyThor suggest putative pathways for protein function. (**a**) Describes an operon in FrontPhageNews that implies that hypothetical proteins gp173 and 174 are likely involved in DNA polymerase assembly. (**b**) Describes an operon in FrontPhageNews that indicates that hypothetical protein gp199 is involved in phage head structure. (**c**) Describes an operon in ChubbyThor that suggests that hypothetical proteins gp43, 44, and 45 are involved in lysis inhibition. (**d**) Describes an operon in ChubbyThor that indicates that hypothetical proteins gp67 and 69 may encode DNA repair proteins.

**Table 1 viruses-16-00289-t001:** Basic genomic characteristics of six Vi01-like phages discovered in the grose lab between 2020 and 2022. The original bacterial host, GenBank accession number, length of genome in base pairs, percent GC composition of the genome, and number of gene products identified are provided.

Phage Name	Bacterial Host	GenBank Accession #	Genome Length (bp)	GC %	Gene Products
AR2819	Salmonella typhimurium	MW021753	156,899	44.97	223
SilasIsHot	Salmonella typhimurium	MW021760	160,559	45.13	227
Sajous1	Salmonella typhimurium	MW021757	157,255	44.86	219
FrontPhageNews	Salmonella typhimurium	MW021754	157,832	44.61	220
ChubbyThor	Shigella boydii	OL615013	159,319	50.1	208
Guerrero	Salmonella typhimurium	OP610151	157,565	44.9	215

**Table 2 viruses-16-00289-t002:** Measurements of select Vi01-like phages. Approximate measurements are provided for representatives from each subcluster. Phages Guerrero and AR2819 were measured using the browser version of program ImageJ 1.54g [95] from electron microscope images in Figure 1b,c, with final measurement composed of the average of three measurements.

Phage	Cluster	Capsid Diameter	Tail Length	Tail Width	Neck	Reference
Vi01	A	89 nm	115 nm	18 nm	*	[73]
Guerrero	A	103 nm	118 nm	23 nm	10 nm	This study
AR2819	A	90 nm	113 nm	28 nm	**	This study
LIMEstone 1	B	91 nm	114 nm	17 nm	20 nm	[20]
UPM2146	C	51 nm	173 nm	10 nm	*	[93]
ϕMAM1	D	90 nm	120 nm	21 nm	11 nm	[69]
3M	D	82 nm	123 nm	18 nm	*	[94]
Bue1	E	79 nm	126 nm	*	*	[35]

* Measurements were not provided. ** EM image is not clear enough to provide accurate measurement.

**Table 3 viruses-16-00289-t003:** Basic genomic characteristics of a representative genome from each of the Vi01-like *Enterobacteriaceae Ackermannviridae* phage subclusters reported in this manuscript. The genome length (in bp), number of current annotated genes, percent genomic GC content (%GC) (for the purpose of at-a-glance comparison of chemical composition of the representative genomes) and GenBank accession number are provided.

Phage	Cluster	Length (bp)	# of Genes	%GC	GenBank Accession	Reference
Vi01	A	157,061	208	45.22	NC_015296	[73]
ChubbyThor	B	159,319	208	50.37	OL615013	[18]
Magnus	C	157,741	217	46.26	MN045230	[19]
3M	D	159,398	203	51.41	NC_048736	(Day, Monson and Salmond, unpublished)
Bue1	E	164,037	176	50.2	NC_048702	[35]

**Table 4 viruses-16-00289-t004:** Average nucleotide identity of one phage from each subcluster of the Vi01-like *Enterobacteriaceae Ackermannviridae*.

Representative	Cluster	Vi01	ChubbyThor	Magnus	3M	Bue1
Vi01	A	1	0.7287	0.7165	0.617	0.6086
ChubbyThor	B	0.7287	1	0.7146	0.6353	0.6233
Magnus	C	0.7165	0.7146	1	0.621	0.6168
3M	D	0.617	0.6353	0.621	1	0.633
Bue1	E	0.6086	0.6233	0.6168	0.633	1

**Table 5 viruses-16-00289-t005:** Core genome for the *Enterobacteriaceae* Vi01-like *Ackermannviridae* Phages with Vi01 gene products as the reference. Protein types are grouped into assembly, DNA/RNA pathways (DNA/RNA), structural, lysis or virulence-related.

Function	Type	Vi01 Gene Product #
RIIA lysis inhibitor	Lysis	1
RIIB lysis inhibitor	Lysis	2
tail fiber	Assembly	3
putative histone like protein	DNA/RNA	4
putative topoisomerase II large subunit	DNA/RNA	5
DNA topoisomerase II small subunit	DNA/RNA	6
putative tRNA processing enzyme	DNA/RNA	7
putative ADP-ribose binding protein	DNA/RNA	8
putative DexA exonuclease	DNA/RNA	9
dCMP deaminase	DNA/RNA	10
membrane-flanked domain protein	Virulence	11
putative head completion protein	Assembly	12
baseplate tail tube cap	Structural	13
baseplate wedge subunit	Structural	14
putative baseplate hub subunit	Structural	15
putative tape measure protein	Structural	16
DNA helicase loader	DNA/RNA	17
putative DNA ligase	DNA/RNA	18
transcriptional regulator	DNA/RNA	19
DNA primase-helicase subunit	DNA/RNA	20
putative RecA protein	DNA/RNA	21
putative dUTP + B23:B63 diphosphatase	DNA/RNA	22
putative dNMP kinase	DNA/RNA	23
putative thymidylate synthase	DNA/RNA	24
putative thymidylate kinase	DNA/RNA	25
putative DNA end protector protein	DNA/RNA	26
putative baseplate tail tube protein	Structural	27
putative ssDNA binding protein	DNA/RNA	28
putative late promoter transcription accessory	DNA/RNA	29
zinc ribbon domain-containing protein	DNA/RNA	30
RuvC-like Holliday junction resolvase	DNA/RNA	31
putative baseplate hub subunit	Structural	32
baseplate hub subunit and tail lysozyme	Structural	33
baseplate wedge subunit	Structural	34
Glutaredoxin	DNA/RNA	35
putative Ribonucleotide-diphosphate reductase beta subunit	DNA/RNA	36
ribonucleoside-diphosphate reductase subunit alpha	DNA/RNA	37
PhoH-like phosphate starvation-inducible gene	Virulence	38
endolysin N-acetylmuramidase	Lysis	39
putative DNA primase	DNA/RNA	40
putative adenylosuccinate synthase	DNA/RNA	41
putative RNA endonuclease	DNA/RNA	42
putative recombination endonuclease subunit	DNA/RNA	43
putative recombination/repair endonuclease subunit	DNA/RNA	44
putative sigma factor for late transcription	DNA/RNA	45
Ribonuclease	DNA/RNA	46
putative ATP-dependent helicase	DNA/RNA	47
putative DNA binding protein	DNA/RNA	48
Rz-like spanin	Virulence	49
putative i-spanin	Virulence	50
putative von Willebrand factor type A domain	Virulence	51
zinc-finger-containing domain protein	DNA/RNA	52
nucleoside triphosphate pyrophosphohydrolase	DNA/RNA	53
RegA-like translation repressor protein	DNA/RNA	54
putative clamp holder for DNA polymerase	DNA/RNA	55
putative clamp loader, small subunit	DNA/RNA	56
putative sliding clamp holder protein	DNA/RNA	57
DNA helicase	DNA/RNA	58
Exonuclease	DNA/RNA	59
UvsY-like recombination mediator	DNA/RNA	60
putative tail completion protein	Assembly	61
Major capsid protein	Structural	62
prohead core scaffold protein	Assembly	63
head maturation protease	Assembly	64
putative prohead core protein	Structural	65
putative portal vertex protein	Structural	66
putative tail tube protein	Structural	67
putative tail sheath protein	Structural	68
terminase large subunit precursor	DNA/RNA	69
putative terminase small subunit	DNA/RNA	70
putative proximal tail sheath stabilizer	Assembly	71
putative neck and head completion protein	Assembly	72
putative neck protein	Structural	73
neck protein	Structural	74
virion structural protein	Structural	75
putative VrlC protein	Virulence	76
putative tail fibers protein	Structural	77
tail spike protein	Structural	78
putative tail fiber	Structural	79
baseplate wedge subunit	Structural	80
baseplate wedge subunit	Structural	81
putative pyridoxal-phosphate dependent enzyme	DNA/RNA	82
putative DNA polymerase	DNA/RNA	83
guanylate kinase	DNA/RNA	84
guanylate kinase	DNA/RNA	85

**Table 6 viruses-16-00289-t006:** Mass spectrometry identifies putative components of the Sajous1 virion proteome. Cesium chloride purified virions were subjected to trypsin digest followed by LC/MS/MS. The number of spectra retrieved for each gene product identified is provided along with their annotated functions, with results sorted by function category (phage structural proteins, phage DNA/RNA processes, cell lysis proteins, phage assembly proteins, miscellaneous proteins, and hypothetical proteins of unknown function. Low spectra counts may indicate contamination from cellular proteins. Raw mass spectrometry data is provided as Appendix A.

**Sajous1 Gene Product**	**Phage Structural Proteins**	**# Spectra Retrieved**
gp157	Major capsid protein	924
gp136	Tail fiber protein	125
gp139	Exo-alpha-sialidase and tail protein	113
gp137	Tail spike protein	96
gp149	Tail sheath protein monomer	89
gp152	Portal vertex protein of the head	89
gp138	Putative tail fiber protein	89
gp55	Phage baseplate wedge protein	80
gp135	Putative tail fiber protein	74
gp53	Putative tape measure protein	52
gp151	Tail tube protein monomer	52
gp83	Putative tail fiber protein	43
gp203	Putative structural protein	39
gp143	Putative neck protein	33
gp115	Putative structural protein	31
gp25	Baseplate tail tube	24
gp134	Head closure	21
gp13	Tail associated lysozyme	20
gp56	T4-like baseplate tail tube cap	17
gp146	Proximal tail completion and sheath stabilization	16
gp119	Putative tail needle knob	16
gp145	Neck and head completion protein	16
gp54	Putative baseplate hub subunit	13
gp55	Baseplate wedge subunit	12
gp116	Phage structural protein	10
gp166	Phage putative structural protein	8
gp168	Tail completion protein	8
gp4	Phage-encoded peptidoglycan binding protein	6
gp12	Putative baseplate wedge protein	6
gp142	putative virion structural protein	2
gp154	Putative prohead core protein	2
**Sajous1 Gene Product**	**Phage DNA/RNA Processes**	**# Retrieved**
gp211	RegB site-specific RNA endonuclease	31
gp192	ParB N-terminal domain containing protein	11
gp24	Single-stranded DNA binding protein	5
gp8	Glutaredoxin	4
gp26	DNA end protector protein	3
gp179	Putative DNA-directed RNA polymerase	3
gp66	QueC-like queuosine biosynthesis protein	1
gp31	Putative thymidylate synthase	1
gp33	Putative dUTP diphosphatase	1
gp174	Sliding clamp loader	1
gp180	Putative DNA-directed RNA polymerase	1
gp185	VWA domain-containing protein	1
**Sajous1 Gene Product**	**Lysis Proteins**	**# Retrieved**
gp87	RIIB protector from prophage-induced early lysis	2
gp88	RllB lysis inhibitor	1
gp59	Membrane protein	1
**Sajous1 Gene Product**	**Phage Assembly Proteins**	**# Retrieved**
gp140	Putative virulence-associated VriC protein	311
gp214	Tail completion protein	196
gp155	Putative prohead protease	13
gp156	Prohead core assembly scaffold	4
gp148	Terminase DNA packaging enzyme large subunit	4
gp147	Terminase DNA packaging enzyme small subunit	1
**Sajous1 Gene Product**	**Miscellaneous Functions**	**# Retrieved**
gp200	SPFH domain band 7 family lipoprotein	3
gp160	Pyruvate: Ferredoxin oxidoreductase	1
gp186	Putative acyl carrier protein	1
**Sajous1 Gene Product**	**Hypothetical proteins**	**# Retrieved**
gp29	Hypothetical protein	10
gp120	Hypothetical protein	4
gp153	Hypothetical protein	2
gp215	Hypothetical protein	32
gp216	Hypothetical protein	20
gp217	Hypothetical protein	10

**Table 7 viruses-16-00289-t007:** Phyre2 analysis of Sajous1 hypothetical proteins identified by mass spectrometry.

Sajous1 Gene Product	Putative Function	Retrieval #	Confidence	% Identity (% Aligned **)
gp29	DNA Polymerase zeta-subunit	10	58.2 *	25 (18)
gp120	Tapasin	4	8.0 *	40 (59)
gp153	Methionine synthase	2	47.2 *	13 (82)
gp215	BiMOP duplicated molybdate-binding domain	32	63	16 (23)
gp216	ATP-dependent DNA helicase, hydrolase	20	27.3 *	43 (27.3)
gp217	Ferritin	10	96.7	22 (96.7)

* Low confidence structural alignments ** The % of the original sequence that aligned with the structure of the proposed template.

**Table 8 viruses-16-00289-t008:** Vi01-like phage host range efficiency of infection results for TonB -and FeoB-deficient *Salmonella* as well as common laboratory *Enterobacteriaceae* strains. All plaque assays were reproducible and were verified in at least two independent experiments with averages provided. ND is no infection detected.

Bacteria Name	AR2819	FrontPhage- News	SilasIsHot	Sajous1	ChubbyThor
*WT *Salmonella* enterica* IR715	1.5 × 10^10^	8.0 × 10^10^	6.9 × 10^10^	8.1 × 10^10^	5.4 × 10^13^
*tonB Salmonella* IR715	1.8 × 10^9^	7.9 × 10^10^	6.6 × 10^10^	7.2 × 10^10^	4.9 × 10^13^
*feoB Salmonella* IR715	1.5 × 10^9^	5.8 × 10^10^	8.1 × 10^9^	8.9 × 10^9^	8.4 × 10^12^
*tonB feoB Salmonella* IR715	1.6 × 10^10^	9.0 × 10^10^	5.9 × 10^10^	8.4 × 10^10^	5.3 × 10^13^
*Salmonella* Typhimurium LT2	3.4 × 10^11^	2.0 × 10^10^	2.6 × 10^10^	6.0 × 10^10^	4.7 × 10^10^
*Citrobacter freundii* ATCC 8090	1.6 × 10^10^	6.0 × 10^9^	2.0 × 10^9^	3.4 × 10^8^	1.7 × 10^9^
*Cronobacter sakazakii* ATCC 29544	ND	ND	ND	ND	ND
*Enterobacter cloacae* ATCC 13047	ND	ND	ND	ND	ND
*Escherichia coli* K12	ND	ND	ND	1.2 × 10^8^	*
*Erwinia amylovora* ATCC 29780	ND	ND	ND	ND	ND
*Serratia marcescens* ATCC 27143	ND	ND	ND	ND	ND
*Shigella boydii* ATCC 9207	ND	ND	ND	ND	7.2 × 10^11^
*Klebsiella pneumoniae* ATCC 10031	ND	ND	ND	ND	ND

* A very clear plaque was observed by spot assay only with the concentrated lysate, indicating possible infection-independent lysis.

**Table 9 viruses-16-00289-t009:** Host range titer results on clinical isolates suggests possible *Salmonella* phage therapy application for Vi01-like phages. Strain *Salmonella* enterica LT2 titer was performed as a reference for the *Escherichia coli* isolates which were assayed on a separate date. All plaque assays were reproducible and were verified in at least two independent experiments with averages provided. ND is no infection detected.

Bacterium *	Strain	AR2819	Front- PhageNews	SilasIsHot	Sajous1	Chubby-Thor
*Salmonella enterica*	#0031	3.1 × 10^12^	1.6 × 10^12^	1.0 × 10^12^	1.5 × 10^11^	7.6 × 10^4^
*Salmonella enterica*	#0409	2.5 × 10^12^	2.7 × 10^8^	1.1 × 10^8^	3.1 × 10^11^	1.2 × 10^9^
*Salmonella heidelberg*	#0404	2.4 × 10^12^	ND	1.80 × 10^11^	3.1 × 10^12^	1.12 × 10^5^
*Salmonella albert*	#0401	ND	ND	ND	ND	ND
*Salmonella cubana*	#0402	ND	ND	ND	ND	ND
*Salmonella infantis*	#0410	ND	ND	ND	ND	ND
*Salmonella senftenberg*	#0405	ND	ND	ND	ND	ND
*Salmonella corvallis*	#0406	ND	ND	ND	ND	ND
*Shigella flexneri*	#0421	ND	ND	ND	ND	ND
*Shigella flexneri*	#0423	ND	ND	ND	ND	ND
*Shigella flexneri*	#0424	ND	ND	ND	ND	ND
*Shigella flexneri*	#0425	ND	ND	ND	ND	ND
*Shigella sonnei*	#0030	ND	ND	ND	ND	ND
*Shigella sonnei*	#0422	ND	ND	ND	1.6 × 10^6^	ND
*Shigella sonnei*	#0426	ND	ND	ND	8.7 × 10^5^	ND
*Citrobacter freundii*	#0021	ND	ND	ND	ND	ND
*Citrobacter freundii*	#0023	ND	ND	ND	ND	ND
*Citrobacter freundii*	#0022	ND	ND	ND	ND	ND
*Citrobacter koseri*	#0024	ND	ND	ND	ND	ND
*Citrobacter koseri*	#0025	ND	ND	ND	ND	ND
*Salmonella enterica*	LT2	1.1 × 10^11^	3.4 × 10^10^	5.4 × 10^10^	8.8 × 10^8^	3.3 × 10^7^
*Escherichia coli* O157:H7	300748	3.6 × 10^10^	1.7 × 10^9^	ND	7.0 × 10^3^	1.5 × 10^9^
*Escherichia coli* O157:H7	300598	1.7 × 10^11^	1.3 × 10^10^	ND	1.4 × 10^7^	8.0 × 10^9^
*Escherichia coli* O157:H7	298559	2.6 × 10^10^	4.0 × 10^9^	ND	1.5 × 10^8^	2.1 × 10^10^
*Escherichia coli* O157:H7	298521	3.7 × 10^11^	2.0 × 10^10^	ND	3.5 × 10^8^	9.4 × 10^8^
*Escherichia coli* O157:H7	290116	8.0 × 10^9^	1.2 × 10^10^	ND	2.4 × 10^7^	4.7 × 10^10^

* Bacterial *Salmonella*, *Shigella* and *Citrobacter* isolates were obtained from the Center for Disease Control and are summarized in Appendix A, while Escherichia coli O157:H7 were obtained from IHC.

**Table 10 viruses-16-00289-t010:** Analysis of tail spike proteins (TSP) of phages AR2819, FrontPhageNews, Guerrero, SilisIsHot, Sajous1 and ChubbyThor. Phage tail fibers were analyzed and classified according to the designations of Sorensen et al. [49]. The phage gene product number is provided, followed by TSP subtype with the phage and its protein used to determine subtype in parenthesis. Subtype hosts and accession numbers are *Escherichia* phage ECML-4 gp89(AFO10352.1) and 190(AFO10351.1), *Salmonella* phage Moolton gp44(AXY85148.1), *Salmonella* phage Maynard gp38 (AGY47760.1), *Escherichia* phage CBA120 gp213(AEM91899.1), *Salmonella* phage Barely orf5 (QIG62071.1), *Salmonella* phage P46FS4 orf5 (QIG62071.1), *Escherichia* phage SA157lWgp4 (AXF39258.1), *Salmonella* phage SFP10 gp161(AEN94256.1), *Salmonella* phage DET7 gp 206 (YP_009140378.1), *Shigella* phage AG3 bp207(YP_003358662.1) and 213(YP_003358665.1).

TSP	AR2819	Front- PhageNews	Guerrero	SilasIsHot	Sajous1	ChubbyThor
TSP1	gp56 **TSP1-20** (ECML-4, gp189)	gp214 **TSP1-20** (ECML-4, gp189)	gp170 **TSP1-20** (ECML-4, gp189)	gp221 **TSP1-18** (SA157lW, gp4)	gp139 **TSP1-20** (ECML-4, gp189)	gp207 **TSP1-24** (AG3, gp207) gp208 **TSP1-22** (P46FS4, gp5) *
TSP2	gp55 **TSP2-1** (ECML-4, gp190)	gp215 **TSP2-1** (ECML-4, gp190)	gp171 **TSP2-1** (ECML-4, gp190)	gp220 **TSP2-5** (Bering, gp7)	gp138 **TSP2-1** (SFP10, gp161)	gp1 **TSP2-1** (SA157lW, gp3)
TSP3	gp5 **TSP3-1** (Moolton gp44)	gp216 **TSP3-1** (Moolton, gp44)	gp172 **TSP3-1** (Moolton, gp44)	gp219 **TSP3-1** (Moolton, gp44)	gp137 **TSP3-1** (Moolton, gp44)	**?**
TSP4	gp53 **TSP4-12** (Maynard, gp38)	gp217 **TSP4-2** (CBA120, gp213)	gp173 **TSP4-9** (DET7, gp206)	gp218 **TSP4-4** (Barely, gp5)	gp136 **TSP4-9** (DET7, gp206)	gp2 **TSP4-14** (AG3, gp213

* All proteins displayed high identity (>80%) to the TSP used for tail fiber typing by NCBI BLASTP with the exception of ChubbyThor gp208 identity to P46FS4 gp5 and gp1 identity to SA157lw gp3 which was only 70.46% over 74% of the protein and 35.79% over 26% of the protein, respectively. ? Indicates that we were unable to identify a TSP3 homolog for ChubbyThor.

## Data Availability

All supporting data is available in the Appendix A.

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
