# Peer review of "Genomic and Proteomic Analysis of Six Vi01-like Phages Reveals Wide Host Range and Multiple Tail Spike Proteins"

_viruses, 2024, doi:10.3390/v16020289_

Round 1
Reviewer 1 Report
Comments and Suggestions for Authors
This study studied six bacteriophages that infect Enterobacteriaceae with genomic and proteomic comparison, mass spectrometry, and host range studies. The authors showed that these phages can infect a broad range of hosts, which could be helpful for Salmonella phage therapy applications. The data presented in the study is solid and lays the groundwork to expand our understanding of the world of phages. However, there are several issues that should be addressed before publication.
Major issues:
Figures 1b and 1c: To support the morphological similarity between the two phages and Vi-O1-like Myoviridae, please provide a detailed table that shows the approximate dimensions of the capsid length, capsid width, tail length, and tail thickness of both phages. Additionally, please provide a figure that compares the two phages with Vi-O1-like Myoviridae based on the images obtained from electron microscopy.
Line 60: Please provide a comprehensive table listing the genomic features of the six phages. ( NCBI accession number, genome size, GC content, protein prediction, and morphotype…)
Minor issues:
Line 100: When you use "ANI" for the first time, please provide its full name.
Line 183: Please provide a brief explanation as to why you compare GC content.
Please revise the manuscript, as there are multiple formatting errors that need to be corrected. Such as line 2, "SixVi01-like", line 28, "1031-32", line 29, "During", and line 408, "table 6".
Please provide the references for the following statements:
- Line 38-39: "Enterobacteriaceae is a group of gram-negative bacteria that was first classified in the 1930s."
- Line 43-44: "These bacteria can cause infections in the urinary tract, intestines, and bloodstream."
- Line 69-70: "Only a few papers have focused on characterizing these phages."
- Line 373: "Tsolis et al. developed Salmonella enterica serovar Typhimurium strains that are deficient in TonB and FeoB."
Author Response
Dear Reviewer 1,
We thank you for your time and suggestions to improve the quality of our manuscript. We agree with your suggestions and have provided point by point response below.
Major issues:
Figures 1b and 1c: To support the morphological similarity between the two phages and Vi-O1-like Myoviridae, please provide a detailed table that shows the approximate dimensions of the capsid length, capsid width, tail length, and tail thickness of both phages. Additionally, please provide a figure that compares the two phages with Vi-O1-like Myoviridae based on the images obtained from electron microscopy.
***Done. See the new Table 2 and discussion (Lines 221-229)
Line 60: Please provide a comprehensive table listing the genomic features of the six phages. ( NCBI accession number, genome size, GC content, protein prediction, and morphotype…)
**Done, See the new Table 3. line 323
Minor issues:
Line 100: When you use "ANI" for the first time, please provide its full name.
**Done, line 123
Line 183: Please provide a brief explanation as to why you compare GC content.
**Done, line 241
Please revise the manuscript, as there are multiple formatting errors that need to be corrected. Such as line 2, "SixVi01-like", line 28, "1031-32", line 29, "During", and line 408, "table 6".
**Done, thank you.
Please provide the references for the following statements:
-Line 38-39: "Enterobacteriaceae is a group of gram-negative bacteria that was first classified in the 1930s."
*done
Line 43-44: "These bacteria can cause infections in the urinary tract, intestines, and bloodstream.
*done
- Line 69-70: "Only a few papers have focused on characterizing these phages."
*done
- Line 373: "Tsolis et al. developed Salmonella enterica serovar Typhimurium strains that are deficient in TonB and FeoB.
- *done
Reviewer 2 Report
Comments and Suggestions for Authors
This paper does a survey of a selection of phages of the Ackermannviridae family. There is one new sequence collated with 150 preexisting sequences. The first main result is to subgroup these using a whole-genome dotplot method. Prototypes of 5 different subgroups are chosen and subjected to a range of analyses, including mass spectroscopy, a scan with Phyre2, some functional inference by gene clustering, and host range analysis. There is also an NJ tree of the MHC protein from a selection of the phages. There are GenBank files for the phages discussed, accessions are given, and the quality of gene annotation, at least for the ones orginating from this group, seems to be relatively high.
This mode of analysis has been well established in the literature. The dotplot analysis is low power but can handle large numbers of phages. It forms one alternative for a variety of first-look methods which are necessary to select a reduced number of phages for more intensive analysis. Normally, when I see this method, it is all phages from a selected host. In this case they are targeting only Vi01-like phages but from a range of hosts in Enterobacteraceae, which is an interesting twist. There is some confusion about how that collection of Vi01-like phages compares to other attempts to define a cluster of Vi01 phages. There is mention of Viunalike viruses, sometimes in italic in the reference list, suggesting that there once was a genus by this name at ICTV. There isn't any more. I sympathize with the authors trying to not use ICTV nomenclature as much as possible, because ICTV nomenclature has been so chaotic and unstable over the years. Nonetheless, you should try to embed some clues that readers can use to collate your taxa with ones that they may know about. The representatives analyzed in the paper include multiple current ICTV genera. They mention that ICTV has created a family around Vi01 named Ackermannviridae. Ackermannviridae has viral genera (and host genera) not found among the representatives named, so it is unclear if this analyzed set includes the same breadth of sequences as Ackermannviridae but with members chosen that just happen to be in Enterobacteracea, or is an unnamed subfamily of Ackermannviridae.
I explored this a bit myself, using the MHC tree in fig.2 as a basis:
%ID in MHC by blastp among Vi01, Tedavirus Aeromnas phage phiA8-29, and Vibrio phage Vap7 is 63%. Those represent the most divergent distances included in Ackermannviridae, or at least they did at some point a few years ago. So I take that as the breadth of Ackermannviridae.
Between Vi01 and clade D identity is 78%. If I was using saturation corrected numbers, that would probably put the root of fig. 2 half way to the root of Ackermannviridae.
Among clades A, B, and C it is about 90%; That should put the root of ABC about half way to the root of (ABCDE).
The root of clade B (Limestone1 vs AV101) is 95%
So this is an unnamed subset of Ackermannviridae. If you use a correction for saturation it will move the more ancient nodes disproportionately back to more ancient times. If corrected the author's Vi01 cluster is probably from a root about half as old as what is currently considered the root of Ackermannviridae. The (D,E) /(A,B,C) node is about twice as old as the (A,B,C) root. As the figure was originally constructed, the (D,E)/(A,B,C) roots should have connected in the center of the figure. Why did that get erased and substituted by the two yellow banana-looking structures? Is it because your ANI values didn't reflect a much deeper split between (D,E) and (A,B,C)? Be aware that ANI doesn't work in that age range. So you overruled a more powerful method with a less powerful method. That is a long explanation for a short revision. If you had put some divergence values on Fig. 2, which the legend actually claims are there, I'd have known exactly what this was at a glance.
I have to say I don't have any confidence in the Phyre2 results. I compared some of them to HHpred and ran one back through Phyre2 to see how the scores of the models you picked compared to the surrounding noise, and I have to say that was pretty disappointing. However, I can't deny that as a survey, Phyre2 might find occasionally find a nugget not noticed by other methods. So I'm not going to say that you shouldn't show it. I noticed that you excluded the ferritin result, which does pan out by hhpred, and any of the numerous proteins where there is clear crystallographic evidence for structural homology among distant phages. You might consider adding a positive control or two to see what a real though highly divergent match looks like with Phyre2. I won't insist on that, but the table really needs pdb numbers.
On that background, these are my major requests for revision, none of which should require any major effort.
1. I assume the Gepard plots were done at the nucleotide level and not as a composite of amino acid sequences. Clarify that and spell out the "default settings" mentioned. In dotplotters I use, I can change signal to noise by orders of magnitude with settings. I don't know how flexible Gepard is, but spell out the settings. Did you use a web server? If so that should be documented in methods.
2. Specify if the MHC tree was made with aligned amino acid or aligned nucleotide sequence. How was the alignment created?
3. On the MHC tree, either create a scale axis giving the Poisson-corrected divergence that the legend claims is there, or state the Poisson-corrected divergence in the legend for at least one node. If your application doesn't give node heights, just branch lengths, then give the branch length of SNUABM-02 and maybe the phage just labeled "38". Something has been done to average the branch lengths and equalize the tip positions in this figure. Was that the timetree option in Mega11, or something else?
4. For the supplemental table with the Phyre2 results (or in the main text if there are Phyre2 results mentioned not found in the supplement), please give the pdb accession of the sequences matched. There are links to papers in the supplement that don't work. Either give the references according to the journal's specifications, or if you give pdb accession, the references are already included at the pdb site. In particular, there are dozens of proteins in human core spliceosome complex. Make it obvious which one are you claiming to match.
5. In the mass spec. tables, # retrieved should be # spectra retrieved, or spectrum count. In the text this was referred to as number of peptides, but it's not. If the same peptide is detected 100 times, that's 100 spectra, and one peptide. You have numbers in those tables much greater than the total number of tryptic peptides in the underlying protein, so what you are actually reporting is # spectra, not # peptides.
Minor issues:
Vi01 is misspelled with either an upper or lower case "O" instead of a zero in a number of places in the manuscript, and at least once with a dash in it. I think it should be Vi01 with a zero; but in any case, please pick one name and stick with it.
"particular" is misspelled in line 294.
SilasIsHot is misspelled SiliasIsHot in lines 59, 128, 145
Author Response
Dear Reviewer 1,
Thank you for the time and consideration of our manuscript, as well as your recommendations for improvement. We have a point by point response to your recommendations.
- I assume the Gepard plots were done at the nucleotide level and not as a composite of amino acid sequences. Clarify that and spell out the "default settings" mentioned. In dotplotters I use, I can change signal to noise by orders of magnitude with settings. I don't know how flexible Gepard is, but spell out the settings. Did you use a web server? If so that should be documented in methods. **We added this to the figure 1 legend. thank you
2. Specify if the MHC tree was made with aligned amino acid or aligned nucleotide sequence. How was the alignment created?
*amino acid and Muscle alignment. We have added the information to the figure legend (Figure 2)
3. On the MHC tree, either create a scale axis giving the Poisson-corrected divergence that the legend claims is there, or state the Poisson-corrected divergence in the legend for at least one node. If your application doesn't give node heights, just branch lengths, then give the branch length of SNUABM-02 and maybe the phage just labeled "38". Something has been done to average the branch lengths and equalize the tip positions in this figure. Was that the timetree option in Mega11, or something else?
*We had not realized that we included the bootstrap tree instead of the original tree which had branch lengths. We actually now supply the original tree on which we overlayed the bootstrap value so that branch lengths are provided (Mega destroys branch lengths when running bootstrap). thank you for catching that error.
4. For the supplemental table with the Phyre2 results (or in the main text if there are Phyre2 results mentioned not found in the supplement), please give the pdb accession of the sequences matched. There are links to papers in the supplement that don't work. Either give the references according to the journal's specifications, or if you give pdb accession, the references are already included at the pdb site. In particular, there are dozens of proteins in human core spliceosome complex. Make it obvious which one are you claiming to match.
** Yes phre2 is a quick analysis so we compared it with HHPRED as you suggest and added the pdb accession and repaired links. (See supplementary Table 3 )
5. In the mass spec. tables, # retrieved should be # spectra retrieved, or spectrum count. In the text this was referred to as number of peptides, but it's not. If the same peptide is detected 100 times, that's 100 spectra, and one peptide. You have numbers in those tables much greater than the total number of tryptic peptides in the underlying protein, so what you are actually reporting is # spectra, not # peptides.
**corrected in the table and text. thank you
Minor issues:
Vi01 is misspelled with either an upper or lower case "O" instead of a zero in a number of places in the manuscript, and at least once with a dash in it. I think it should be Vi01 with a zero; but in any case, please pick one name and stick with it.
**corrected. thank you
"particular" is misspelled in line 294.
**corrected. thank you
SilasIsHot is misspelled SiliasIsHot in lines 59, 128, 145
**corrected. thank you
Round 2
Reviewer 1 Report
Comments and Suggestions for Authors
The authors have addressed my concerns. I support the publication of this manuscript.